# Parameter-efficient Multi-Task and Multi-Domain Learning using Factorized Tensor Networks

## Abstract

Multi-task and multi-domain learning methods seek to learn multiple tasks/domains, jointly or one after another, using a single unified network. The primary challenge and opportunity lie in leveraging shared information across these tasks and domains to enhance the efficiency of the unified network. The efficiency can be in terms of accuracy, storage cost, computation, or sample complexity. In this paper, we introduce a factorized tensor network (FTN) designed to achieve accuracy comparable to that of independent single-task or single-domain networks, while introducing a minimal number of additional parameters. The FTN approach entails incorporating task- or domain-specific low-rank tensor factors into a shared frozen network derived from a source model. This strategy allows for adaptation to numerous target domains and tasks without encountering catastrophic forgetting. Furthermore, FTN requires a significantly smaller number of task-specific parameters compared to existing methods. We performed experiments on widely used multi-domain and multi-task datasets. We show the experiments on convolutional-based architecture with different backbones and on transformer-based architecture. Our findings indicate that FTN attains similar accuracy as single-task or single-domain methods while using only a fraction of additional parameters per task.

## 1 Introduction

The primary objective in multi-task learning (MTL) is to train a single model that learns multiple related tasks, either jointly or sequentially. Multi-domain learning (MDL) aims to achieve the same learning objective across multiple domains. MTL and MDL techniques seek to improve overall performance by leveraging shared information across multiple tasks and domains. On the other hand, single-task or single-domain learning does not have that opportunity. Likewise, the storage and computational cost associated with single-task/domain models quickly grows as the number of tasks/domains increases. In contrast, MTL and MDL methods can use the same network resources for multiple tasks/domains, which keeps the overall computational and storage cost small Mallya et al. (2018); Berriel et al. (2019); Wallingford et al. (2022); Rebuffi et al. (2018); Mancini et al. (2018); Sun et al. (2020); Kanakis et al. (2020); Maninis et al. (2019).

In general, MTL and MDL can have different input/output configurations, but we model them as task/domain-specific network representation problems. Let us represent a network for MTL or MDL as the following general function:

$$\mathbf{y}_t = \mathbf{F}_t(\mathbf{x}) \equiv \mathbf{F}(\mathbf{x}; \mathcal{W}_t, h_t), \tag{1}$$

where $\mathbf{F}_t$ represents a function for task/domain $t$ that maps input $\mathbf{x}$ to output $\mathbf{y}_t$. We further assume that $\mathbf{F}$ represents a network with a fixed architecture and $\mathcal{W}_t$ and $h_t$ represent the parameters for task/domain-specific feature extraction and classification/inference heads, respectively. The function in equation 1 can represent the network for specific task/domain $t$ using the respective $\mathcal{W}_t, h_t$. In the case of MTL, with $T$ tasks, we can have $T$ outputs $\mathbf{y}_1, \ldots, \mathbf{y}_T$ for a given input $\mathbf{x}$. In the case of MDL, we usually have a single output for a given input, conditioned on the domain $t$. Our main goal is to learn the $\mathcal{W}_t, h_t$ for all $t$ that maximize the performance of MTL/MDL with minimal computation and memory overhead compared to single-task/domain learning.

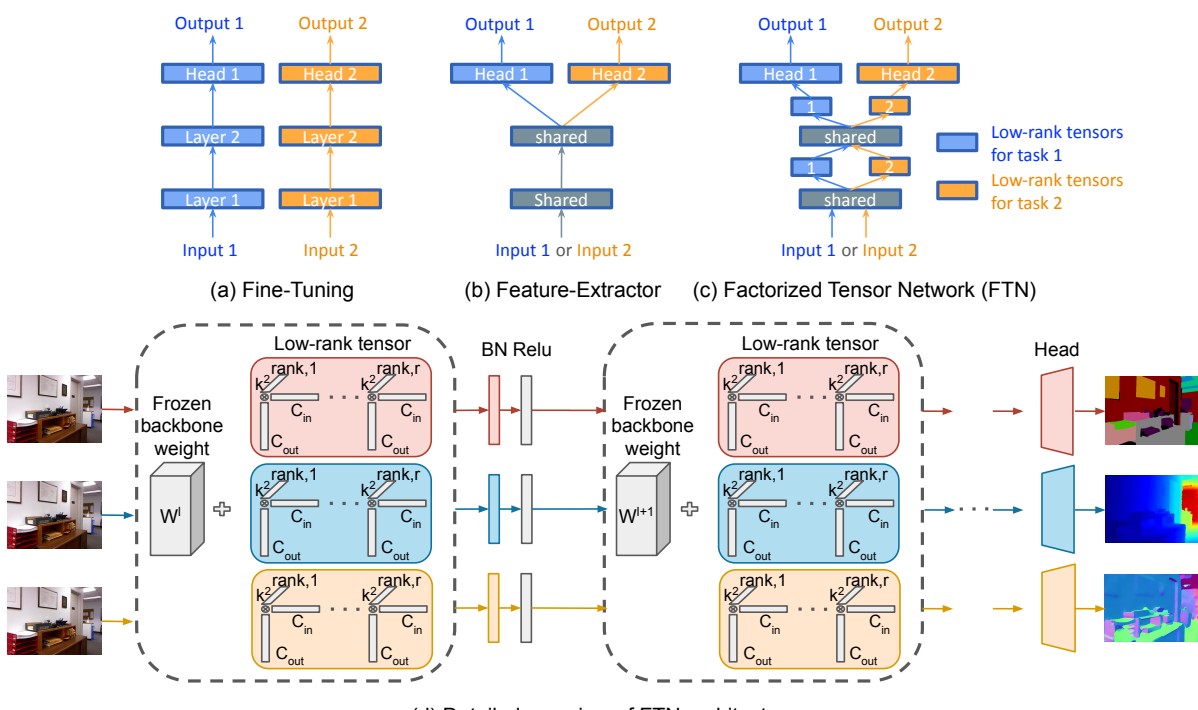

(a) Fine-Tuning   (b) Feature-Extractor   (c) Factorized Tensor Network (FTN)

(d) Detailed overview of FTN architecture.

**Figure 1:** Overview of different MTL/MDL approaches and our proposed method. (a) Fine-Tuning trains entire network per task/domain. (b) Feature-Extractor trains a backbone network shared by all tasks/domains with task/domain-specific heads. (c) Our proposed method, Factorized Tensor Network (FTN), adapts to a new task/domain by adding low-rank factors to shared layers. (d) Detailed overview of FTN. A single network adapted to three downstream vision tasks (segmentation, depth, and surface normal estimation) by adding task-specific low-rank tensors ($\Delta \mathcal{W}_t$). Task/domain-specific blocks are shown in same colors.

Figure 1(a),(b),(c) illustrate three typical approaches for MTL/MDL. First, we can start with a pre-trained network and fine-tune all the parameters ($\mathcal{W}_t$) to learn a target task/domain, as shown in Figure 1(a). Fine-Tuning approaches can transfer some knowledge from the pretrained network to the target task/domain, but they effectively use an independent network for every task/domain Mallya et al. (2018); Wallingford et al. (2022); Tzeng et al. (2017); Venkateswara et al. (2017); Mustafa et al. (2021); Kolesnikov et al. (2020). Second, we can reduce the parameter and computation complexity by using a completely shared Feature-Extractor (i.e., $\mathcal{W}_t = \mathcal{W}_{\text{shared}}$ for all $t$) and learning task/domain-specific heads as last layers, as shown in Figure 1(b). While such approaches reduce the number of parameters, they often result in poor overall performance because of limited network capacity and interference among features for different tasks/domains Mallya et al. (2018); Berriel et al. (2019); Wallingford et al. (2022); Zhang et al. (2020). Third, we can divide the network into shared and task/domain-specific parameters or pathways, as shown in Figure 1(c). Such an approach can increase the network capacity, provide interference-free paths for task/domain-specific feature extraction, and enable knowledge sharing across the tasks/domains. In recent years, a number of such methods have been proposed for MTL/MDL Wallingford et al. (2022); Kanakis et al. (2020); Misra et al. (2016); Ruder et al. (2019); Gao et al. (2019); Strezoski et al. (2019); Liang et al. (2018); Gao et al. (2020); Yu et al. (2020). While existing methods can provide performance comparable to single-task/domain learning, they require a significantly large number of additional parameters.

In this paper, we propose a new parameter-efficient method to divide network into shared and task/domain-specific parameters using a factorized tensor network (FTN). In particular, our method learns task/domain-specific low-rank tensor factors and normalization layers. An illustration of our proposed method is shown in Figure 1(d), where we represent network parameters as $\mathcal{W}_t = \mathcal{W}_{\text{shared}} + \Delta \mathcal{W}_t$, where $\Delta \mathcal{W}_t$ is a low-rank tensor. Furthermore, we also learn task/domain-specific normalization parameters. We demonstrate the effectiveness of our method using different MTL and MDL datasets. Our method can achieve accuracy comparable to a single-task/domain network with a small number of additional parameters. A prior work,

TAPS Wallingford et al. (2022), differentially learns which layers of a pre-trained network to adapt for a downstream task/domain by learning an indicator function. The network uses adapted weights instead of pre-trained weights if the indicator score is above a certain threshold. This typically involves adapting high-parameterized layers closer to the classifier/head, which uses significantly more parameters than our FTN method. Through experiments, we show that TAPS uses approximately 2 to 4 times more parameters than FTN across various datasets and tasks. Existing parameter-efficient MTL/MDL methods Mallya et al. (2018); Mallya & Lazebnik (2018); Li et al. (2016) introduce small task/domain-specific parameters while others Zhang et al. (2020); Guo et al. (2019) add many parameters to boost the performance irrespective of the task complexity. In our work, we demonstrate the flexibility of FTNs by selecting the rank according to the complexity of the task. Other approaches like RCM Kanakis et al. (2020) adapt incrementally to new tasks by reparameterizing the convolutional layer into task-shared and task-specific parameters. However, unlike FTN this architecture shows limitations in adapting based on the complexity of the tasks and performs subpar along performance and parameters.

**Contributions.** The main contributions of this paper can be summarized as follows.

- We propose a new method for MTL and MDL, called factorized tensor networks (FTN), that adds task/domain-specific low-rank tensors to shared weights.
- We demonstrate that by using a fraction of additional parameters per task/domain, FTNs can achieve similar performance as the single-task/domain methods.
- Our proposed FTNs can be viewed as a plug-in module that can be added to any pretrained network and layer. We have shown this by extending FTNs to transformer-based architectures.
- We performed empirical analysis to show that the FTNs enable flexibility by allowing us to vary the rank of the task-specific tensors according to the problem complexity.

**Limitations.** In our experiments, we used a fixed rank for each layer. In principle, we can adaptively select the rank for different layers to further reduce the parameters. MTL/MDL models often suffer from task interference or negative transfer learning when trained jointly with multiple conflicting tasks. Our method can have similar drawbacks as we did not investigate which tasks/domains should be learned jointly. A shared backbone requires a single forward pass for all tasks, while our proposed FTN would require as many forward passes as the number of tasks. Branched and tree-structures can enable different tasks to share several layers and reduce latency.

## 2 Related Work

**Multi-task learning (MTL)** methods commonly leverage shared and task-specific layers in a unified network to solve related tasks Misra et al. (2016); Gao et al. (2019); Liu et al. (2019); Bruggemann et al. (2020); Xu et al. (2018); Zhang et al. (2018); Vandenhende et al. (2020); Zhang et al. (2022c;a). These methods learn shared and task-specific representation through their respective modules. Optimization-based methods Chen et al. (2018c); Kendall et al. (2018); Chen et al. (2020) devise a principled way to evaluate gradients and losses in multi-task settings. Branched and tree-structured MTL methods Bruggemann et al. (2020); Guo et al. (2020); Zhang et al. (2022b) enable different tasks to share branches along a tree structure for several layers. Multiple tasks can share computations and features in any layer only if they belong to the same branch in all the preceding layers. Kanakis et al. (2020); Maninis et al. (2019) proposed MTL networks that incrementally learn new tasks. ASTMT Maninis et al. (2019) proposed a network that emphasizes or suppresses features depending on the task at hand. RCM Kanakis et al. (2020) reparameterizes the convolutional layer into non-trainable and task-specific trainable modules. We compare our proposed method with these incrementally learned networks. Adashare Sun et al. (2020) is another related work in MTL that jointly learns multiple tasks. It learns task-specific policies and network pathways Jang et al. (2017).

**Multi-domain learning (MDL)** focuses on adapting one network to multiple unseen domains or tasks. MDL setup trains models on task-specific modules built upon the frozen backbone network. This setup helps MDL networks avoid negative transfer learning or catastrophic forgetting, which is common among

multi-task learning methods. The work by Rebuffi et al. (2017; 2018) introduces the task-specific parameters called residual adapters. The architecture introduces these adapters as a series or parallel connection on the backbone for a downstream task. Inspired by pruning techniques, Packnet Mallya & Lazebnik (2018) learns on multiple domains sequentially on a single task to decrease the overhead storage, which comes at the cost of performance. Similarly, the Piggyback Mallya et al. (2018) method uses binary masks as the module for task-specific parameters. These masks are applied to the weights of the backbone to adapt them to new domains. To extend this work, WTPB Mancini et al. (2018) uses the affine transformations of the binary mask on their backbone to extend the flexibility for better learning. BA$^2$ Berriel et al. (2019) proposed a budget-constrained MDL network that selects the feature channels in the convolutional layer. It gives a parameter-efficient network by dropping the feature channels based on budget but at the cost of performance. Zhao et al. (2021) paper learns the adapter modules and the plug-in architecture of the modules using NAS. Spot-Tune Guo et al. (2019) learns a policy network, which decides whether to pass each image through Fine-Tuning or pretrained networks. It neglects the parameter efficiency factor and emphasises more on performance. TAPS Wallingford et al. (2022) adaptively learns to change a small number of layers in a pretrained network for the downstream task.

**Domain adaptation and transfer learning.** The work in this field usually focuses on learning a network from a given source domain to a closely related target domain. The target domains under this kind of learning typically have the same category of classes as source domains Tzeng et al. (2017). Due to this, it benefits from exploiting the labels of source domains to learn about multiple related target domainsVenkateswara et al. (2017); Li et al. (2021). Some work has a slight domain shift between source and target data, like different camera views Saenko et al. (2010). At the same time, recent papers have worked on significant domain shifts like converting targets into sketch or art domains Venkateswara et al. (2017); Zhao et al. (2017). Transfer learning is related to MDL or domain adaptation but focuses on better generalizing target tasks Mustafa et al. (2021); Kolesnikov et al. (2020); Dosovitskiy et al. (2021). Most of the work in this field uses the popular ImageNet as a source dataset to learn feature representation and learn to transfer to target datasets. The method proposed in Yang et al. (2022) uses a pretrained (multi-task) teacher network and decomposes it into multiple task/knowledge-specific factor networks that are disentangled from one another. This factorization leads to sub-networks that can be fine-tuned to downstream tasks, but they rely on knowledge transfer from a teacher network that is pretrained for multiple tasks. Modular deep learning methods Pfeiffer et al. (2023) focus on transfer learning by avoiding negative task interference and having parameter-efficient modules.

**Factorization methods in MDL/MTL.** The method in Yang & Hospedales (2015) proposed a unified framework for MTL/MDL using semantic descriptors, without focusing on parameter-efficient adaptation. Yang & Hospedales (2017) performs MTL/MDL by factorizing each layer in the network after incorporating task-specific information along a separate dimension. Both the networks in Yang & Hospedales (2015) and Yang & Hospedales (2017) require retraining from scratch for new tasks/domains. In contrast, FTN can incrementally learn low-rank factors to add new tasks/domains. Chen et al. (2018b) proposed a new parameter-efficient network to replace residual networks by incorporating factorized tensors. The results in Chen et al. (2018b) are limited to learning single-task networks, where the network is only compressed by up to $\sim 60\%$. In Bulat et al. (2020), the authors proposed a network for MDL using Tucker decomposition.

**Transformer-based methods in MDL/MTL.** LoRA Hu et al. (2021) is a low-rank adaptation method proposed for large language models, which freezes the pre-trained weights of the model and learns low-rank updates for each transformer layer. It updates weight matrices for query and value in every attention layer. Similarly, KAdaptation He et al. (2023) proposes a parameter-efficient adaptation method for vision transformers. It represents the updates of MHSA layers using the summation of Kronecker products between shared parameters and low-rank task-specific parameters. We compared both of these methods and have shown that FTN outperforms along the number of parameters. Scaling and shifting your features (SSF) Lian et al. (2022) is another transformer method for parameter-efficient adaptation that applies element-wise multiplication and addition to tokens after different operations. SSF, in principle, is similar to fine-tuning the Batch Normalization layer in convolutional layers, which has scaling and shifting trainable parameters. FTN trains the Batch Normalization layers and has the same effect as scaling and shifting features when adapting

to new tasks. Ye & Xu (2022) proposed inverted-pyramid multi-task transformer, performs cross-task interaction among spatial features of different tasks in a global context. The DeMT Xu et al. (2023) proposes a deformable mixer encoder and task-aware transformer decoder. The proposed encoder leverages channel mixing and deformable convolution operation for informative features while the transformer decoder captures the task interaction. Our method, FTN, shares some high-level similarities with other parameter-efficient adaptation methods such as LoRA, as both approaches aim to introduce low-rank factors to adapt networks for multiple tasks and domains. Our method is a natural extension to higher-order tensors, and we have demonstrated its effectiveness across both transformer and convolutional network architectures. In addition, our method adds parameter and performance efficiency compared to related method, as demonstrated by our experiments.

In summary, our proposed method (FTN) offers a parameter-efficient approach to achieve performance comparable to or better than existing adaptation methods by utilizing a fraction of additional parameters. Our primary design consideration was to achieve efficient adaptation, enabling incremental learning with additive factors. To achieve parameter efficiency, we introduce a small number of trainable parameters through low-rank factorization applicable to both convolutional and transformer-based networks. We utilize frozen and trainable task-specific parameters to support incremental learning without forgetting prior knowledge.

## 3 Technical Details

**Notations.** In this paper, we denote scalars, vectors, matrices and tensors by $w$, $\mathbf{w}$, $W$, and $\mathbf{W}$, respectively. The collective set of tensors (network weights) is denoted as $\mathcal{W}$.

### 3.1 FTN applied to Convolutional layers

In our proposed method, we use task/domain-specific low-rank tensors to adapt every convolutional layer of a pretrained backbone network to new tasks and domains. Let us assume the backbone network has $L$ convolutional layers that are shared across all task/domains. We represent the shared network weights as $\mathcal{W}_{\text{shared}} = \{\mathbf{W}_1, \ldots, \mathbf{W}_L\}$ and the low-rank network updates for task/domain $t$ as $\Delta\mathcal{W}_t = \{\Delta\mathbf{W}_{1,t}, \ldots, \Delta\mathbf{W}_{L,t}\}$. To compute features for task/domain $t$, we update weights at every layer as $\mathcal{W}_{\text{shared}} + \Delta\mathcal{W}_t = \{\mathbf{W}_1 + \Delta\mathbf{W}_{1,t}, \ldots, \mathbf{W}_L + \Delta\mathbf{W}_{L,t}\}$.

To keep our notations simple, let us only consider $l$th convolutional layer that has $k \times k$ filters, $C_{in}$ channels for input feature tensor, and $C_{out}$ channels for output feature tensor. We represent the corresponding $\mathbf{W}_l$ as a tensor of size $k^2 \times C_{in} \times C_{out}$. We represent the low-rank tensor update as a summation of $R$ rank-1 tensors as

$$\Delta\mathbf{W}_{l,t} = \sum_{r=1}^{R} \mathbf{w}_{1,t}^r \otimes \mathbf{w}_{2,t}^r \otimes \mathbf{w}_{3,t}^r, \tag{2}$$

where $\mathbf{w}_{1,t}^r, \mathbf{w}_{2,t}^r, \mathbf{w}_{3,t}^r$ represent vectors of length $k^2, C_{in}, C_{out}$, respectively, and $\otimes$ represents the Kronecker product.

Apart from low-rank tensor update, we also optimize over Batch Normalization layers (BN) for each task/domain Ioffe & Szegedy (2015); Pham et al. (2022). The BN layer learns two vectors $\Gamma$ and $\beta$, each of length $C_{out}$. The BN operation along $C_{out}$ dimension can be defined as element-wise multiplication and addition:

$$\text{BN}_{\Gamma,\beta}(u) = \Gamma\left(\frac{u - \mathbb{E}[u]}{\sqrt{\text{Var}[u] + \epsilon}}\right) + \beta. \tag{3}$$

We represent the output of $l$th convolutional layer for task/domain $t$ as

$$\mathbf{Z}_{l,t} = \text{BN}_{\Gamma_t,\beta_t}(\text{conv}(\mathbf{W}_l + \Delta\mathbf{W}_{l,t}, \mathbf{Y}_{l-1,t})), \tag{4}$$

where $\mathbf{Y}_{l-1,t}$ represents the input tensor and $\mathbf{Z}_{l,t}$ represents the output tensor for $l$th layer. In our proposed FTN, we learn the task/domain-specific factors $\{\mathbf{w}_{1,t}^r, \mathbf{w}_{2,t}^r, \mathbf{w}_{3,t}^r\}_{r=1}^R$, and $\Gamma_t$, and $\beta_t$ for every layer in the backbone network.

In the FTN method, rank $R$ for $\Delta\mathbf{W}$ plays an important role in defining the expressivity of the adapted network. We can define a complex $\Delta\mathbf{W}$ by increasing the rank $R$ of the low-rank tensor and taking their linear combination. Our experiments showed that this has resulted in a significant performance gain.

**Initialization.** To establish a favorable starting point, we adopt a strategy that minimizes substantial modifications to the frozen backbone network weights during the initialization of the task-specific parameter layers. To achieve this, we initialize each parameter layer from the Xavier uniform distribution Glorot & Bengio (2010), thereby generating $\Delta\mathbf{W}$ values close to 0 before their addition to the frozen weights. This approach ensures the initial point of our proposed network closely matches the pretrained network closely.

To acquire an effective initialization for our backbone network, we leverage the pretrained weights obtained from ImageNet. We aim to establish a robust and capable feature extractor for our specific task by incorporating these pretrained weights into our backbone network.

**Number of parameters.** In a Fine-Tuning setup with $T$ tasks/domains, the total number of required parameters at convolutional layer $l$ can be calculated as $T \cdot (k^2 \times C_{in} \times C_{out})$. Whereas using our proposed FTNs, the total number of frozen backbone ($\mathbf{W}_l$) and low-rank R tensor ($\Delta\mathbf{W}_{l,t}$) parameters are given by $(C_{out} \times C_{in} \times k^2) + T \cdot R \cdot (C_{out} + C_{in} + k^2)$. In our results section, we have shown that the absolute number of parameters required by our method is a fraction of what the Fine-Tuning counterpart needs.

**Effect of Batch Normalization.** In our experiment section, under the 'FC and BN only' setup, we have shown that having task-specific Batch Normalization layers in the backbone network significantly affects the performance of a downstream task/domain. For all the experiments with our proposed approach, we include Batch Normalization layers as task-specific along with low-rank tensors and classification/decoder layer.

## 3.2 FTN applied to Transformers

The Vision Transformer (ViT) architecture Dosovitskiy et al. (2020) consists a series of MLP, normalization, and Multi-Head Self-Attention (MHSA) blocks. The MHSA blocks perform $n$ parallel attention mechanisms on sets of Key $K$, Query $Q$, and Value $V$ matrices. Each of these matrices has dimensions of $S \times d_{model}$, where $d_{model}$ represents the embedding dimension of the transformer, and $S$ is the sequence length. The $i$-th output head ($H_i$) of the $n$ parallel attention blocks is computed as

$$H_i = \text{SA}(Q\text{W}_i^Q, K\text{W}_i^K, V\text{W}_i^V), \tag{5}$$

where $\text{SA}(\cdot)$ represents the self-attention mechanism, $\text{W}_i^K, \text{W}_i^Q, \text{W}_i^V \in \mathbb{R}^{d_{model} \times d}$ represent the projection weights for the key, query, and value matrices, respectively, and $d = d_{model}/n$. The heads $H_i$ are then combined using a projection matrix $\text{W}_o \in \mathbf{R}^{d_{model} \times d_{model}}$ to result in the output of the MHSA block as

$$\text{MHSA}(H_1, \ldots, H_n) = \text{Concat}(H_1, \ldots, H_n) \cdot \text{W}_o. \tag{6}$$

Following the adaptation procedure in He et al. (2023), we apply our proposed factorization technique to the weights in the MHSA block. We introduce two methods for applying low-rank tensors to the attention weights:
**Adapting query and value weights.** Our first proposed method, *FTN (Query and Value)*, adds the low-rank tensor factors to the query $\text{W}^Q$ and value $\text{W}^V$ weights. These weights can be represented as three-dimensional tensors of size $d_{model} \times d \times n$. Using equation 2, we can define and learn low-rank updates $\Delta\mathbf{W_q}$ and $\Delta\mathbf{W_v}$ for the query and value weights, respectively.
**Adapting output weights.** Our second method, *FTN (Output projection)*, adds low-rank factors, $\Delta\mathbf{W}_o$, to the output projection weights $\mathbf{W}_o \in \mathbf{R}^{d_{model} \times d \times n}$. Similar to the previous low-rank updates, the updates to the output weights defined following equation 2.

**Initialization.** We initialize each low-rank factor by sampling from a Gaussian distribution with $\mu = 0$ and $\sigma = 0.05$. This ensures near-zero initialization, closely matching the pretrained network.

**Number of parameters.** The total number of parameters needed for $R$ low-rank tensors and $L$ MHSA blocks in FTN (Query and Value) is $2LR(d_{model}+d+n)$. FTN (Output Projection) requires only $LR(d_{model}+$

**Table 1:** Number of parameters and top-1% accuracy for baseline methods, comparative methods, and FTN with varying ranks on the five domains of the ImageNet-to-Sketch benchmark experiments. Additionally, the mean top-1% of each method across all domains is shown. The 'Params' column gives the number of parameters used as a multiplier of those for the Feature-Extractor method, along with the absolute number of parameters required in parentheses.

| Methods | Params (Abs) | Flowers | Wikiart | Sketch | Cars | CUB | mean |
|---|---|---|---|---|---|---|---|
| Fine-Tuning | 6× (141M) | 95.69 | 78.42 | 81.02 | 91.44 | 83.37 | 85.98 |
| Feature-Extractor | 1× (23.5M) | 89.57 | 57.7 | 57.07 | 54.01 | 67.20 | 65.11 |
| FC and BN only | 1.001× (23.52M) | 94.39 | 70.62 | 79.15 | 85.20 | 78.68 | 81.60 |
| Piggyback Mallya et al. (2018) | 6× [2.25×] (141M) | 94.76 | 71.33 | 79.91 | 89.62 | 81.59 | 83.44 |
| Packnet → Mallya & Lazebnik (2018) | [1.60×] (37.6M) | 93 | 69.4 | 76.20 | 86.10 | 80.40 | 81.02 |
| Packnet ← Mallya & Lazebnik (2018) | [1.60×] (37.6M) | 90.60 | 70.3 | 78.7 | 80.0 | 71.4 | 78.2 |
| Spot-Tune Guo et al. (2019) | 7× [7×] (164.5M) | 96.34 | 75.77 | 80.2 | 92.4 | 84.03 | 85.74 |
| WTPB Mancini et al. (2018) | 6× [2.25×] (141M) | 96.50 | 74.8 | 80.2 | 91.5 | 82.6 | 85.12 |
| BA$^2$ Berriel et al. (2019) | 3.8× [1.71×] (89.3M) | 95.74 | 72.32 | 79.28 | 92.14 | 81.19 | 84.13 |
| TAPS Wallingford et al. (2022) | 4.12× (96.82M) | 96.68 | 76.94 | 80.74 | 89.76 | 82.65 | 85.35 |
| **FTN, R=1** | **1.004×** (23.95M) | 94.79 | 73.03 | 78.62 | 86.85 | 80.86 | 82.83 |
| **FTN, R=50** | 1.53× (36.02M) | 96.42 | 78.01 | 80.6 | 90.83 | 82.96 | **85.76** |

$d + n$) to add a similar number of factors. These additional parameters are significantly fewer than the parameters required for fully fine-tuning the four attention weights, which equals $4Ld_{model}^2$. When compared to other parameter-efficient adaptation methods such as LoRA Hu et al. (2021) and KAdaptation He et al. (2023), our methods show superior parameter efficiency. The primary distinction is in the method of weight factorization and decomposition. In LoRA, to introduce rank $R$ factors in the query and value weight matrices, $4LRd_{model}$ parameters are required. Our approach begins with a three-dimensional representation of the attention weights, sized $d_{model} \times d \times n$. We chose this approach because it allows us to exploit the relationship between the attention heads, further improving parameter efficiency. Moreover, we have explored different types of updates within the self-attention mechanism and proposed two variants of our FTN (*Query and Value* and *Output projection*). ~~LoRA requires $4LRd_{model}$ parameters to apply low-rank factors to query and value weight matrices.~~ KAdaptation requires $2LRd_{model} + K^3$ additional parameters for each weight, where $K$ represents a design parameter. SSF Lian et al. (2022) requires $mLd_{model}$, where $m$ is the number of SSF modules in each transformer layer. In Table 3, we report the exact number of parameters and demonstrate that our proposed method, *FTN (Output Projection)*, has the best parameter efficiency.

## 4 Experiments and Results

We evaluated the performance of our proposed FTN on several MTL/MDL datasets. We performed experiments for **1. Multi-domain classification** on convolution and transformer-based networks, and **2. Multi-task dense prediction**. For each set of benchmarks, we reported the performance of FTN with different rank increments and compared the results with those from existing methods. All experiments are run on a single NVIDIA GeForce RTX 2080 Ti GPU with 12GB memory.

### 4.1 Multi-domain classification

#### 4.1.1 Convolution-based networks

**Datasets.** We use two MTL/MDL classification-based benchmark datasets. First, ImageNet-to-Sketch, which contains five different domains: Flowers Nilsback & Zisserman (2008), Cars Krause et al. (2013), Sketch Eitz et al. (2012), Caltech-UCSD Birds (CUBs) Wah et al. (2011), and WikiArt Saleh & Elgammal (2016), with different classes. Second, DomainNet Peng et al. (2019), which contains six domains: Clipart, Sketch, Painting (Paint), Quickdraw (Quick), Inforgraph (Info), and Real, with each domain containing an equal 345 classes. The datasets are prepared using augmentation techniques as adopted by Wallingford et al. (2022).

**Training details.** For each benchmark, we report the performance of FTN for various choices for ranks, along with several benchmark-specific comparative and baseline methods. The backbone weights are pre-trained from ImageNet, using ResNet-50 He et al. (2016) for the ImageNet-to-Sketch benchmarks, and ResNet-34 on the DomainNet benchmarks to keep the same setting as Wallingford et al. (2022). On ImageNet-to-Sketch we run FTNs for ranks, $R \in \{1, 5, 10, 15, 20, 25, 50\}$ and on DomainNet dataset for ranks, $R \in \{1, 5, 10, 20, 30, 40\}$. In the supplementary material, we provide the hyperparameter details to train FTN.

**Results.** We report the top-1% accuracy for each domain and the mean accuracy across all domains for each collection of benchmark experiments. We also report the number of frozen and learnable parameters in the backbone network. Table 1 compares the FTN method with other methods in terms of accuracy and number of parameters. FTN outperforms every other method while using 36.02 million parameters in the backbone with rank-50 updates for all domains. The mean accuracy performance is better than other methods and is close to Spot-Tune Guo et al. (2019), which requires nearly 165M parameters. On the Wikiart dataset, we outperform the top-1 accuracy with other baseline methods. The performance of baseline methods is taken from TAPS Wallingford et al. (2022) since we are running the experiments under the same settings.

**Table 2:** Performance of different methods with resnet-34 backbone on DomainNet dataset. Top-1% accuracy is shown on different domains with different methods along with the number of parameters.

| Methods | Params | Clipart | Sketch | Paint | Quick | Info | Real | mean |
|---|---|---|---|---|---|---|---|---|
| Fine-Tuning | 6× | 74.26 | 67.33 | 67.11 | 72.43 | 40.11 | 80.36 | 66.93 |
| Feature-Extractor | 1× | 60.94 | 50.03 | 60.22 | 54.01 | 26.19 | 76.79 | 54.69 |
| FC and BN only | 1.004× | 70.24 | 61.10 | 64.22 | 63.09 | 34.76 | 78.61 | 62.00 |
| Adashare Sun et al. (2020) | 5.73× | 74.45 | 64.15 | 65.74 | 68.15 | 34.11 | 79.39 | 64.33 |
| TAPS Wallingford et al. (2022) | 4.90× | 74.85 | 66.66 | 67.28 | 71.79 | 38.21 | 80.28 | **66.51** |
| **FTN, R=1** | **1.008×** | 70.73 | 62.69 | 65.08 | 64.81 | 35.78 | 79.12 | 63.03 |
| **FTN, R=40** | 1.18× | 74.2 | 65.67 | 67.14 | 71.00 | 39.10 | 80.64 | 66.29 |

Table 2 shows the results on the DomainNet dataset, which we compare with TAPS Wallingford et al. (2022) and Adashare Sun et al. (2020). Again, using FTN, we significantly outperform comparison methods along the required parameters (rank-40 needs 25.22 million parameters only). Also, FTN rank-40 attains better top-1% accuracy on the Infograph and Real domain, while it attains similar performance on all other domains. On DomainNet with resnet-34 and Imagenet-to-Sketch with resnet-50 backbone, the rank-1 low-rank tensors require only 16,291 and 49,204 parameters per task, respectively. We have shown additional experiments on this dataset under a joint optimization setup in the supplementary material.

**Analysis on rank.** We create low-rank tensors ($\Delta W$) as a summation of $R$ rank-1 tensors. We hypothesize that increasing $R$ increases the expressive power of low-rank tensors. Our experiments confirm this hypothesis, where increasing the rank improves the performance, enabling more challenging task/domain adaptation. Figure 2 shows the accuracy vs. ranks plot, where we observe a trend of performance improvement as we increase the rank from 1 to 50 on the ImageNet-to-Sketch and from 1 to 40 on the DomainNet dataset. In addition, we observe that some domains do not require high ranks. Particularly, the Flowers and Cars domains attain good accuracy at ranks 20 and 15, respectively. We can argue that, unlike prior works Guo et al. (2019); Li et al. (2016), which consume the same task-specific module for easy and complex tasks, we can provide different flexibility to each task. Also, we can add as many different tasks as we want by adding independent low-rank factors for each task (with a sufficiently large rank). In supplementary material, we present a heatmap that shows the adaption of the low-rank tensor at every layer upon increasing the rank.

### 4.1.2 Transformer-based networks

We compared our FTN method with several domain adaptation techniques for supervised image classification. Our task is to adapt a pretrained 12-layer ViT-B-224/32 (CLIP) model obtained from He et al. (2023) to new domains.

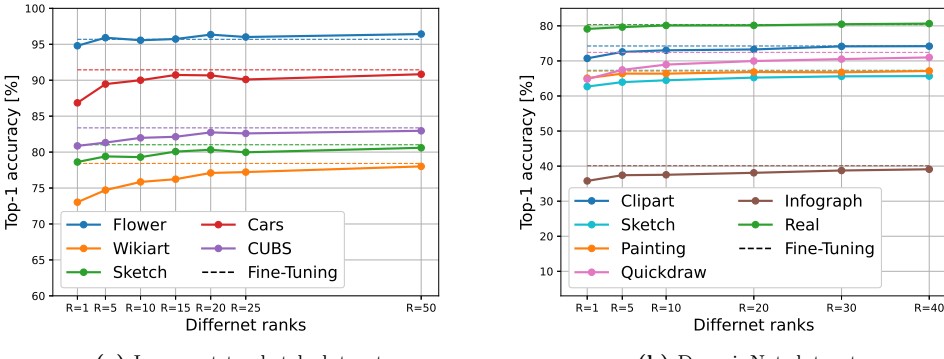

**(a)** Imagenet-to-sketch dataset        **(b)** DomainNet dataset

**Figure 2: Accuracy vs Low-ranks:** We show the top-1% accuracy against the different low-ranks used in our method for different domains. We start with an 'only BN' setup where without any low-rank we keep the Batch Normalization layers as task-specific. Then we show the performance improvement through our approach upon increasing the rank, R.

**Table 3:** We compared performance across five datasets in terms of accuracy and total parameters. FTN (O) uses low-rank factors for output projection weights, while FTN (Q&V) applies them to query and value weights. Note that the parameters mentioned exclude task-specific heads, and $5 \times (439.5M)$ denotes a fivefold increase in base network parameters, $5 \times 87.9M$.

| Method | # total params | # additional params | CIFAR10 | CIFAR100 | DTD | STL10 | FER2013 | mean |
|---|---|---|---|---|---|---|---|---|
| Fine-tuning | $5 \times$ (439.5M) | $5 \times$ 87.9M | **97.7** | **85.4** | **79.0** | **99.7** | **69.8** | **86.3** |
| Feature extractor | $1 \times$ (87.9M) | - | 94.8 | 80.1 | 75.4 | 98.4 | 67.3 | 83.2 |
| LoRA Hu et al. (2021) | $1.008 \times$ (88.6M) | $5 \times$147.2K | 95.1 | 78.1 | 78.1 | 99.2 | 67.7 | 83.6 |
| KAdaptation He et al. (2023) | $1.005 \times$ (88.3M) | $5 \times$80.7K | 95.9 | 84.8 | 78.1 | 99.2 | 69.0 | 85.4 |
| **FTN (Q & V)** | $1.005 \times$ (88.3M) | $5 \times$ 81.0K | 95.8 | 83.4 | 77.1 | 98.7 | 68.5 | 84.7 |
| **FTN (O)** | $1.002 \times$ (88.1M) | $5 \times$40.5K | 96.6 | 84.3 | 76.0 | 98.6 | 69.5 | 85.0 |

**Datasets.** We conducted experiments on the CIFAR10 Krizhevsky et al. (2009), CIFAR100 Krizhevsky et al. (2009), DTD Cimpoi et al. (2014), FER2013 Goodfellow et al. (2013), and STL10 Coates et al. (2011) classification datasets, using the official dataset splits.

**Training details.** For all experiments, we set the rank to $R = 4$. We followed a similar hyper-parameter tuning procedure and implementation as outlined in He et al. (2023), which utilizes grid-search to obtain the optimal learning rate for each dataset. We found that $5 \times 10^{-6}$ was the optimal learning rate. Following the approach in Hu et al. (2021), we scaled the low-rank factors by $\frac{\alpha}{R}$, where $\alpha$ is a hyper-parameter, and $R$ is the number of low-rank factors. We set $\alpha = 10$ and $\alpha = 100$ for FTN (Query and Value) and FTN (Output projection), respectively. We used a batch size of 64 and trained for 100 epochs.

**Results.** In Table 3, we present the classification accuracy and the total number of parameters for our proposed FTN methods, along with related model adaptation methods. Results for Fine-tuning, Feature extractor (Linear-probing), LoRA Hu et al. (2021), and KAdaptation He et al. (2023) are obtained from He et al. (2023). The first proposed method, FTN (query and value), surpasses LoRA in terms of average performance and requires fewer additional parameters. FTN (query and value) requires a comparable number of parameters to KAdaptation and performance is 0.8% lower. In contrast, FTN (output projection) requires approximately half as many additional parameters as KAdaptation but achieves comparable performance.

## 4.2 Multi-task dense prediction

**Dataset.** The widely-used NYUD dataset Silberman et al. (2012) with 795 training and 654 testing images of indoor scenes is used for dense prediction experiments in multi-task learning. The dataset contains four tasks: edge detection (Edge), semantic segmentation (SemSeg), surface normals estimation (Normals), and depth estimation (Depth). We follow the same data-augmentation technique as used by Kanakis et al. (2020).

**Metrics.** On the tasks of the NYUD dataset, we report mean intersection over union for semantic segmentation, mean error for surface normal estimation, optimal dataset F-measure Martin et al. (2004) for edge

detection, and root mean squared error for depth estimation. We also report the number of parameters used in the backbone for each method.

**Training details.** ResNet-18 is used as the backbone network, and DeepLabv3+ Chen et al. (2018a) as the decoder architecture. The Fine-Tuning and Feature-Extractor experiments are implemented in the same way as in the classification-based experiments above. We showed experiments for FTNs with $R \in \{1, 10, 20, 30\}$. Further details are in the supplementary material.

**Results.** Table 4 shows the performance of FTN with various ranks and of other baseline comparison methods for dense prediction tasks on the NYUD dataset. We observe performance improvement by increasing flexibility through higher rank. FTN with rank-30 performs better than all comparison methods and utilizes the least number of parameters. Also, on the Depth and Edge task we can attain good performance by using only rank-20. We take the performance of baseline comparison methods from the RCM paper Kanakis et al. (2020) as we run our experiments under the same setting.

**Table 4:** Dense prediction performance on NYUD dataset using ResNet-18 backbone with DeepLabv3+ decoder. The proposed FTN approach with $R = \{1, 10, 20, 30\}$ and other methods. The best performing method in bold.

| Methods | Params | Semseg↑ | Depth↓ | Normals↓ | Edge↑ |
|---|---|---|---|---|---|
| Single Task | 4× | 35.34 | 0.56 | 22.20 | 73.5 |
| Decoder only | 1× | 24.84 | 0.71 | 28.56 | 71.3 |
| Decoder + BN only | 1.002× | 29.26 | 0.61 | 24.82 | 71.3 |
| ASTMT (R-18) Maninis et al. (2019) | 1.25× | 30.69 | 0.60 | 23.94 | 68.60 |
| ASTMT (R-26+SE) Maninis et al. (2019) | 2.00× | 30.07 | 0.63 | 24.32 | 73.50 |
| Series RA Rebuffi et al. (2018) | 1.56× | 31.87 | 0.60 | 23.35 | 67.56 |
| Parallel RA Rebuffi et al. (2018) | 1.50× | 32.13 | 0.59 | 23.20 | 68.02 |
| RCM Kanakis et al. (2020) | 1.56× | 34.20 | 0.57 | 22.41 | 68.44 |
| **FTN, R=1** | **1.005×** | 29.83 | 0.60 | 23.56 | 72.7 |
| **FTN, R=10** | 1.03× | 33.66 | 0.57 | 22.15 | 73.5 |
| **FTN, R=20** | 1.06× | 34.06 | **0.55** | 21.84 | **73.9** |
| **FTN, R=30** | 1.09× | **35.46** | 0.56 | **21.78** | 73.8 |

## 5    Conclusion

We have proposed a simple, parameter-efficient, architecture-agnostic, and easy-to-implement FTN method that adapts to new unseen domains/tasks using low-rank task-specific tensors. Our work shows that FTN requires the least number of parameters compared to other baseline methods in MDL/MTL experiments and attains better or comparable performance. We can adapt the backbone network in a flexible manner by adjusting the rank according to the complexity of the domain/task. We conducted experiments with different convolutional backbones and transformer architectures for various datasets to demonstrate that FTN outperforms existing methods.

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
