# OpenReview forum: "Parameter-efficient Multi-Task and Multi-Domain Learning using Factorized Tensor Networks"
_TMLR — Rejected by TMLR_

### Review · Reviewer_dhwp · 2024-05-24

**Summary Of Contributions:**

>> Context:

This paper addresses an important technical problem regarding how to adapt a pre-trained model to a new task or domain while requiring as little compute and few additional parameters as possible. It is indeed a relevant question for many practical applications these days. The paper is focused on classification tasks for computer vision.

>> Summary:

- The paper proposes to use low-rank updates for two different architectures: convolutional networks and Transformers. In the case of CNNs, the updates are applied to convolutional filters and batch norms. In the case of Transformers, updates are applied either to query and value parameter matrices, or to the projection matrix within the attention layer. Every task (or, say, downstream dataset) requires finetuning these low rank matrices.

- The paper offers an interesting comparison for increasing values of R (the rank of the update). We see that higher rank does better (as expected) but performance plateaus fairly quickly (good news, I guess).

- Experiments on ResNets (CNN) and VIT (Transformer) suggest FTN outperforms most alternatives in terms of performance while --at the same time-- requiring a significantly lower parameter count overhead.

**Audience:**

Yes

**Claims And Evidence:**

Yes

**Requested Changes:**

>> Questions:

- What's the exact difference between Transformer FTN and LoRA? I think the paper could do a bit of a better job at explaining the technical difference as, from the outside, both methods seem fairly similar. Is FTN just a different way to decompose the update matrix?

- For LoRA (W_upd = BA), one of the parameter matrices (B) is initialized to be identical to zero. Accordingly, the initial checkpoint produces the exact same outputs as the pretrained model. In this work, the algorithm randomly initialize all the new parameters (to small values). I wonder what's the effect of this, and whether authors have ablated other types of initializations closer to that of LoRA.

**Strengths And Weaknesses:**

I'm not sure what's different between this and previous low-rank adapter approaches, especially LoRA (even though the related work section is thorough). Overall, it seems novelty is limited, while experiments are well explained and interesting.

---

> ### Author Response · Authors · 2024-06-10
> **Response to Reviewer dhwp**
>
> **Re: Difference between Transformer FTN and LoRA**
>
> - FTN shares some high-level similarities with other parameter-efficient adaptation methods such as LoRA, as both approaches aim to introduce low-rank factors to adapt networks for multiple tasks and domains. Originally, LoRA was proposed to update weight matrices using two low-rank factors. Our method is a natural extension to higher-order tensors, and we have demonstrated its effectiveness across both transformer and convolutional network architectures. In addition, our method adds parameter and performance efficiency compared to LoRA, as demonstrated by our experiments.
>
> - Addressing your question about Transformer FTN, the primary distinction is in the method of weight factorization and decomposition.  In LoRA, suppose the network has $L$ transformer layers with word embedding $d_{\text{model}}$. To introduce rank $R$ factors in the query and value weight matrices, $4LRd_{\text{model}}$ parameters are required. Our approach begins with a three-dimensional representation of the attention weights, sized $d_{\text{model}} \times d_h \times n$, where $n$ is the number of parallel attention blocks and $d_{\text{model}}$ equals $d_h \times n$. It requires $2LR(d_{model}+d+n)$ parameters to update the Query and Value attention weights. We chose this approach because it allows us to exploit the relationship between the attention heads, further improving parameter efficiency. Moreover, we have explored different types of updates within the self-attention mechanism and proposed two variants of our FTN (Q&V and O). We have expanded the differences between LoRA and FTN in the related works and method section (Revised changes are on page 4 and 7, highlighted in blue).
>
> **Re: Initialization effect LoRA vs FTN.**
>
> - Regarding the outputs of initial checkpoints, since our initialization method assigns each low-rank factor value that is near zero (obtained from a zero-mean $\sigma = 0.05$ Gaussian distribution), the added weight is very close to zero. As a result, we expect similar output to the initial checkpoint. To verify this, we conducted an experiment where we used a pre-trained 12-layer ViT-B-224/32 (CLIP) backbone network to perform inference on four datasets and compared it to the initial checkpoint (without any fine-tuning). The datasets we used are CIFAR-10, CIFAR-100, FER-2013, and DTD. As shown in Table Re3, the difference in performance (accuracy) from the initial checkpoint is negligible, proving our assumption that both initialization methods indeed result in similar initial performance.
>
> Table Re3: Initial checkpoint results for different initialization methods on four different datasets (validation accuracy)
>
> | Initialization method  | CIFAR 10 | CIFAR 100 | FER 2013 | DTD  |
> |------------------------|----------|-----------|----------|------|
> | LoRA (B=0)             | 88.52    | 1.18      | 42.27    | 1.28 |
> | FTN (random)           | 88.01    | 1.13      | 42.38    | 1.17 |
>
> - **Regarding the ablations of initialization types closer to LoRA**: For multi-head attention, we use three low-rank factors, which we will refer to as A, B, and C, that are multiplied together, to form a dense tensor or the attention weights. If we naively initialize B = 0 and C = 0, similar to LoRA, our gradient-based optimization will get stuck. This is because the gradient of the loss with respect to A, B, or C will involve a product with either B or C, which are zero, resulting in zero gradients for any low-rank factors. While setting only one factor to zero initially is an option, it raises the question of which factor to choose and requires further ablation studies. To avoid this issue, we opted for a simpler and more intuitive initialization scheme that maintains the outputs of the pretrained weights at the start without explicitly zeroing out specific factors.

---

### Review · Reviewer_wxxw · 2024-05-24

**Summary Of Contributions:**

This paper proposed Factorized Tensor Network to boost multi-task learning and multi-domain learning. FTN attains similar accuracy as single-task or single-domain methods while using only a fraction of additional parameters per task.

**Audience:**

No

**Claims And Evidence:**

Yes

**Requested Changes:**

- The idea is not new. The overall contribution is incremental.
- In my opinion, an important application for multi-task learning is to get better generalization ability by reusing knowledge in several tasks that are similar and can share knowledge with other tasks. It is not satisfactory to only get similar accuracy with single-task methods.
- The experiments are not very comprehensive. What about multi-task applications that are commonly used in the read-world? For example, object detection?

**Strengths And Weaknesses:**

Strengths:
+ The targeted problem is important.
+ The paper is good to follow.

Weaknesses:
- The idea is not new. The overall contribution is incremental.
- In my opinion, an important application for multi-task learning is to get better generalization ability by reusing knowledge in several tasks that are similar and can share knowledge with other tasks. It is not satisfactory to only get similar accuracy with single-task methods.
- The experiments are not very comprehensive. What about multi-task applications that are commonly used in the read-world? For example, object detection?

---

> ### Author Response · Authors · 2024-06-10
> **Response to Reviewer wxxw**
>
> **Re: Contribution is incremental.**
>
> We are performing tensor decomposition, which has been explored previously by some works. However, our method, FTN, proposes a simple and different way of decomposition that has proven effective in reducing the number of parameters. FTN achieves accuracy that is better than or similar to other works, such as RCM, LoRA, and KAdaptation while using fewer parameters.
>
> **Re: Reusing knowledge / sharing knowledge**
>
> - We agree that good knowledge sharing can improve the performance of MTL/MDL. Nevertheless, MTL/MDL methods can use a pretrained (frozen) network and add parameter-efficient adaptations (also called incremental MTL/MDL) or train a shared network using all the domains/tasks along with adaptations (also called joint MTL/MDL). For instance, LoRA method adds task-specific low-rank matrices to pretrained (frozen) weights; the Visual Decathlon dataset paper (Rebuffi et al. NeurIPS 2017) adds task-specific 1x1 filters and batch norm layers in a pretrained network; RCM paper also promotes incremental MTL approach.
>
> - Two justifications often provided in favor of incremental MTL/MDL vs joint MTL/MDL are as follows. (1) Jointly training multiple tasks/domains can cause interference/negative transfer learning and cause performance degradation; (2) Incremental learning provides flexibility for adding new tasks/domains later when the corresponding data/labels become available (see additional discussion in TAPS and RCM papers).
>
> - We included one experiment with joint MDL on DomainNet in Table 3 of supplementary, where FTN with rank-1 updates and joint MDL achieves better performance than incremental MDL with rank-40 (in Table 2, main paper). We have performed another experiment (Table Re2) with joint MTL on the NYUv2 dataset, where we have shared the backbone among 4 different tasks, (joint-multitask setup in the Table Re2). We can observe performance improvement over Decoder only setup and for depth estimation we have achieved performance similar to single-task. We have further experimented with FTN, Rank=1 using the joint-multitask backbone weights and adapting task-specific low-rank adapters. FTN with joint-multitask backbone improves the performance compared with FTN, Rank=1 imagenet backbone.
>
> Table Re2: Performance on the NYUv2 dataset. Joint-multitask row shows the performance on each task with backbone trained on all tasks jointly. FTN, Rank=1 w/ joint-backbone row shows the performance with frozen backbone initialized from joint-multitask weights instead of imagenet weights and the low-rank adapters and heads are trained incrementally for each task.
>
> |                                 | Semseg↑ | Depth↓ | Normals↓ |   |
> |---------------------------------|---------|--------|----------|---|
> | single-task                     | 35.34   | 0.56   | 22.20    |   |
> | Decoder only                    | 24.84   | 0.71   | 28.56    |   |
> | FTN, Rank=1 w/ imagenet weights | 29.83   | 0.60   | 23.56    |   |
> | joint-multitask                 | 30.59   | 0.56   | 23.37    |   |
> | FTN, Rank=1 w/ joint-backbone   | 31.03   | 0.55   | 23.41    |   |
>
> **Re: Experiments on real-world multi-task applications.**
>
> - Thank you for your feedback regarding the experiments, particularly on multi-task applications that are commonly used in the real-world. We have conducted experiments on multiple classification domains and different dense prediction tasks to show the effectiveness of our method. FTN can easily be transferred to different tasks/domains by adding the task/domain specific low-rank adapters in the backbone architecture.
>
> - We selected the datasets to be consistent with the cohort of previous works done in incremental MTL/MDL and to have a direct comparison with those methods. In addition, the multi-task dataset NYUv2 contains semantic segmentation, normals, saliency, and edge detection as different tasks that have further downstream applications in autonomous vehicles, and robot and drone navigation that are useful in the real-world.

---

### Review · Reviewer_onsF · 2024-05-27

**Summary Of Contributions:**

This paper introduces a novel Factorized Tensor Network (FTN) that can be integrated into CNNs and Transformers for multi-task and multi-domain learning. The FTN aims to reduce model redundancy in these scenarios. Experimental results demonstrate that using the FTN allows a unified network to achieve accuracy comparable to independent single-task or single-domain networks, while requiring only a minimal increase in parameters.

**Audience:**

Yes

**Broader Impact Concerns:**

No.

**Claims And Evidence:**

Yes

**Requested Changes:**

1. Conduct a comprehensive review of previous FTN works, providing more detailed information.

2. Carefully adjust the learning settings to highlight the practical advantages of the proposed approach.

3. Design the baseline experiments meticulously to ensure solid empirical validation.

4. Cite the works of the comparison methods when presenting the result tables.

**Strengths And Weaknesses:**

## Strengths

1. The paper proposes a novel Factorized Tensor Network (FTN) for parameter-efficient multi-task and multi-domain learning with a clear motivation. The addition of task/domain-specific low-rank tensors to shared weights is logically sound.

2. The FTN is versatile, capable of being integrated into both CNNs and Transformers, demonstrating flexibility across different network architectures.

3. Experimental results on multi-task and multi-domain datasets show that the FTN achieves comparable accuracy to baseline and previous methods, with only a minimal increase in additional parameters.

4. The paper is well-organized and presented in a manner that is easy to follow.

## Weaknesses

1. When introducing the FTN, it is crucial to clearly differentiate it from previous works such as those by Wallingford et al. and Kanakis et al. etc. This distinction helps readers understand the novelty and specific benefits of the proposed FTN, rather than just listing related works, which might confuse readers unfamiliar with the area.

2. From the result tables, it appears that the main benefit of the proposed FTN is parameter reduction. The fine-tuning method consistently yields the best results. Is there potential to further improve accuracy in multi-task and multi-domain learning compared to single-task fine-tuning? The goal of multi-task learning is to enhance the performance of each task; otherwise, it is unnecessary. Thus, the learning settings should be carefully adjusted, as storage cost is cheap while accuracy and inference speed are more critical in practice.

3. The experiment in Table 2 is not very convincing. Note that all six tasks are the same across different domains in the DomainNet dataset, making individual heads for each task unnecessary. Using a unified head might yield better results in multi-domain learning compared to the Fine Tuning method. Therefore, the baseline methods should be carefully designed to ensure solid empirical experiments.

4. There are typos in Figure 1(c). For example, [shared]->[1 1]->[shared]->[2 2] should likely be [shared]->[1 2]->[shared]->[1 2]?

---

> ### Author Response · Authors · 2024-06-10
> **Response to Reviewer onsF**
>
> **Re: A comprehensive review of previous works**
>
> Thank you for the suggestion. In the revised main paper, we have expanded the introduction section to highlight the differences compared to previous works, providing a better understanding of the novelty and benefits offered by FTN. (Revised changes are on page 3, highlighted in blue.)
>
> **Re: Performance improvement beyond single-task and adjust the learning settings to highlight the practical advantages**
>
> - The performance of multi-task and multi-domain learning can be further improved compared to single-task learning under a joint-learning setup (sharing information between the tasks) [1, 2, 3]. However, it is critical to evaluate which tasks or domains help or interfere when trained jointly. RCM paper [Kanakis, et al.] has conducted a study to demonstrate task-interference by using Representation Similarity Analysis (RSA), which shows limited correlation of the gradients among the different tasks. To avoid such limitations, this paper primarily focuses on incremental learning. Incremental multi-task learning (MTL) and multi-domain learning (MDL) involve adding task-specific parameters to a frozen or pre-trained network. Our experiments are designed to incrementally adapt to new domains/tasks by learning task-specific low-rank adapters. Another advantage of incremental MTL/MDL is its efficiency in adapting to new domains/tasks compared to a joint MTL/MDL setup. For example, let's assume we initially trained on N domains jointly and then a new domain is introduced. Adapting to all N+1 domains under a joint setup becomes sub-optimal and requires data from all previous domains. In contrast, our method can learn the new domain without any prior data. Moreover, the primary advantage of our method, FTN, lies in parameter reduction while maintaining accuracy that is better than or similar to that of previous works.
>
> - **Performance beyond the single-task learning** - On the DomainNet dataset, which contains common objects in 6 different domains (i.e. each object in a category is in the form of clipart images, real photos, sketches, and so on), we showed an additional experiment in Table 3 of the supplementary section. As observed in Table 2 of the main paper (Fine-Tuning row) and Table 3 of the supplementary section (FTN, R=1), sharing information between tasks enhances performance beyond single-task learning in some domains.
>
> **Re: Meticulously designing baseline experiments, experiment with a unified head on the DomainNet dataset.**
>
> - Thank you for the feedback. We conducted an experiment with the DomainNet dataset, which consists of six distinct domains. We combined all six domains and trained a network that uses a single backbone and classifier. Table Re1 shows significant performance improvements in sketch and painting using the unified classifier. However, accuracy is lower in clipart, quickgraph, and real compared to independent models for each domain. This experiment indicates that even with similar tasks, a unified head performs subpar, suggesting Fine Tuning and Feature extractor methods are better baselines. We designed three baseline setups: Fine-Tuning, where each domain is trained independently; Feature Extractor, with frozen backbones and independent heads for each domain; and FC and BN only, where only the heads and batch normalization layers are optimized to avoid task/domain interference. We combined the experiment with a unified head in the revised version of Table 3 of the supplementary section.
>
> Table Re1: Performance on the DomainNet dataset using a ResNet-34 backbone. The first row shows the performance of models independently trained for each domain. The second row shows the performance for each domain after training a single network with a single classifier.
> |                                                      | clipart | sketch | painting | quick | info  | real  | mean  |
> |------------------------------------------------------|---------|--------|----------|-------|-------|-------|-------|
> | Fine-tuning / individual models                       | 74.26   | 67.33  | 67.11    | 72.43 | 40.11 | 80.36 | 66.93 |
> | Unified head      | 33.74   | 73.97  | 73.13    | 54.75 | 39.49 | 63.64 | 56.45 |
>
> - As mentioned in the previous response, we showed on the DomainNet dataset (under the supplementary section) that the performance improves under the backbone sharing setup (with independent heads) compared to the individually trained network.
>
> **Re: Cite the works of the comparison methods.**
>
> In the revised main paper, we have cited the works of comparison methods in the tables.
>
> **Re: Typos in Figure 1(c).**
>
> We have corrected the figure accordingly.
>
> [1] Brüggemann, David, et al. "Exploring relational context for multi-task dense prediction."
>
> [2] Vandenhende, Simon, et al. "Mti-net: Multi-scale task interaction networks for multi-task learning."
>
> [3] Misra, Ishan, et al. "Cross-stitch networks for multi-task learning."

---

> > ### Comment · Reviewer_onsF · 2024-06-12
> >
> > I would like to ask if all your experiments are conducted based on an incremental learning setup? I did not see specific details regarding the incremental learning experiments.

---

> ### Author Response · Authors · 2024-06-12
> **details of the incremental learning experiments**
>
> Thank you for your question.
>
> All the experiments reported in the main text are performed under incremental learning setup. We freeze the weights of the pretrained backbone networks and learn the batch normalization layers, low-rank tensors, and the classifier heads for each specific task or domain individually.
>
> For the multi-domain classification experiments, we used pretrained convolutional and transformer networks.
> In Sec. 4.1.1, the backbone weights are pretrained from ImageNet, using ResNet-50 He et al. (2016) for the ImageNet-to-Sketch benchmarks, and ResNet-34 on the DomainNet benchmarks to keep the same setting as Wallingford et al. (2022).
> In Sec. 4.1.2, we adapt a pretrained 12-layer ViT-B-224/32 (CLIP) model obtained from He et al. (2023) to new domains.
>
> For the multi-task dense prediction experiments, ResNet-18 is used as the backbone network (also pretrained with ImageNet dataset). We freeze the ResNet-18 convolutional layers and learn only the batch normalization layers and low-rank tensors, along with task-specific decoders with DeepLabv3+ (Chen et al., 2018a) architecture. We follow the same incremental learning setup that was used in the RCM paper (Kanakis, Menelaos, et al., 2020).
>
> In the supplementary material, we included one experiment in A.4, where we train the backbone network for all the domains jointly. We then freeze the network and learn additional domain-specific parameters. The main takeaway of Table S3 is that a network jointly trained on multiple domains with task-specific batch normalization (or rank-1 update) can achieve better performance than AdaShare and TAPS.
>
> Our experiments follow the setup described in the technical details section, and we have included the pretrained network details in the experiments section. We will be happy to further clarify and highlight our use of the incremental learning framework.

---

> > ### Comment · Reviewer_onsF · 2024-06-12
> >
> > The incremental learning setup in the SCM paper is only detailed for the NYUD dataset. What about the specific settings for other datasets, such as DomainNet? It is important to clarify the order in which different tasks or domains are run.
> >
> > Moreover, if this paper focuses on incremental learning rather than multi-task learning, it is crucial to clearly state this unified point. As it stands, it is difficult to discern whether the paper is addressing incremental learning. And if so, more SOTA works about incremental learning should be discussed.

---

> > > ### Author Response · Authors · 2024-06-12
> > > **Incremental learning setup for different tasks/domains**
> > >
> > > Thank you again for your question.
> > >
> > > The settings for DomainNet are consistent with those described in the TAPS paper Wallingford et al. (2022). We would like to clarify something regarding the "incremental" MDL/MTL framework mentioned in our paper and in related works, such as RCM and TAPS. The domains/tasks are learned independently, rather than sequentially, using a shared backbone network with task/domain-specific batch normalization layers and low-rank tensor factors. This means that the tasks can be learned in any order or even in parallel, without influencing the performance. We did not design this setup, we are simply following the setup proposed in other papers for a fair comparison.
> > >
> > > We will be happy to further clarify this point in the paper.

---

> > > > ### Comment · Reviewer_onsF · 2024-06-13
> > > >
> > > > Thanks for your reply. I understand the setup now. It appears that it is neither traditional multi-task learning nor traditional incremental learning. I suggest adjusting the title or content of the paper to reflect this, as it may otherwise be easily misunderstood. Additionally, regarding this learning setup, I still feel that its significance is not very clear, since its performance does not surpass that of single-task fine-tuning. The only apparent advantage seems to be the reduced memory overhead.

---

> > > > > ### Author Response · Authors · 2024-06-13
> > > > >
> > > > > Glad to offer the clarification.
> > > > >
> > > > > As we mentioned, we are following the setup proposed in other papers for a fair comparison. We clearly described our framework in the abstract and the main paper. We will be happy to further clarify the learning setup.
> > > > >
> > > > > We agree with you that the main advantage of our method is the reduced memory overhead. We demonstrate that by using a fraction of additional parameters per task/domain, FTNs can achieve similar performance as the single-task/domain methods. Learning a single task/domain by fine tuning the entire network per task/domain is expected to be the upper limit in this framework, and we do not claim to surpass that.

---

> > > > > > ### Comment · Reviewer_onsF · 2024-06-13
> > > > > >
> > > > > > However, it seems that the LoRA method can achieve better results than full fine-tuning, based on the results reported in their paper. Why are these results not replicated in your paper?

---

> > > > > > > ### Author Response · Authors · 2024-06-13
> > > > > > > **Full fine-tuning vs LoRA**
> > > > > > >
> > > > > > > Thanks for raising this question. While the results reported in the original LoRa paper show competitive performance, sometimes surpassing full-tuning, the experiments and evaluation metrics differ significantly from our setup. Our experiments focus on vision classification and dense prediction tasks using the ViT backbone. We utilized a publicly available implementation of LoRA applied to the ViT architecture from KAdaptation (He et al., 2023) to reproduce the results. We set the rank to 4 and performed a grid search to determine the optimal learning rate and $\alpha$ values. In related work such as [1,2], it is shown that LoRA with R=4 does not always outperform full network fine-tuning on related vision tasks.
> > > > > > >
> > > > > > > [1] Agiza, Ahmed, Marina Neseem, and Sherief Reda. “MTLoRA: Low-Rank Adaptation Approach for Efficient Multi-Task Learning.” Proceedings of the IEEE/CVF Conference on Computer Vision and Pattern Recognition. 2024.
> > > > > > >
> > > > > > > [2] Tang, Yiwen, et al. “Point-PEFT: Parameter-efficient fine-tuning for 3D pre-trained models.” Proceedings of the AAAI Conference on Artificial Intelligence. Vol. 38. No. 6. 2024.

---

> > > > > > > > ### Comment · Reviewer_onsF · 2024-06-14
> > > > > > > >
> > > > > > > > Alright, but it seems that the MTLoRA method can also achieve better results than full fine-tuning. This method appears to be similar to your setup. How does your method compare to MTLoRA?

---

> > > > > > > > > ### Author Response · Authors · 2024-06-14
> > > > > > > > >
> > > > > > > > > MTLoRA method does not seem to achieve better results than full fine-tuning for all the tasks. The results are mixed (e.g., Human Parts performance is better for single task full fine-tuning). The paper does not present any experiment for multi-domain adaptation, which was the main topic we were discussing above.
> > > > > > > > >
> > > > > > > > > We recently found the MTLoRA paper. A fair comparison of our method with MTLoRA will require a significant effort, which seems unreasonable for the following reason: MTLoRA is a CVPR 2024 paper (CVPR will start next week) that appeared on ArXiv in March 2024. We submitted our paper in Feb 2024. We have to draw a line somewhere to stop the comparisons.
> > > > > > > > >
> > > > > > > > > We will be happy to cite the MTLoRA paper, but we cannot keep adding new experiments and comparisons.

---

> > > > > > > > > > ### Comment · Reviewer_onsF · 2024-06-14
> > > > > > > > > >
> > > > > > > > > > Thank you for your patient response. I now have one final question. Regarding the technical aspects, it seems that the KAdaptation (He et al., 2023) method is not fundamentally different from the method presented in this paper. Could you elaborate on the additional innovations and advantages of the FTN method compared to the KAdaptation method?

---

> > > > > > > > > > > ### Author Response · Authors · 2024-06-14
> > > > > > > > > > >
> > > > > > > > > > > Thanks for your question. We are happy to discuss the differences between FTN and KAdaptation. Both methods aim to achieve an efficient and effective factorization of network parameters, but they differ in how the weight tensors or matrices are defined and decomposed.
> > > > > > > > > > >
> > > > > > > > > > > KAdaptation represents updates to the Multi-Head Self-Attention (MHSA) layers using the summation of Kronecker products, $\sum_i A_i \otimes B_i$. The number of parameters required by KAdaptation with $L$ number of layers and rank $r$ is $2Lrd_{model} + K^3$, where $K$ is a design parameter, and $d_{model}$ is the embedding dimension. KAdaptation reduces the number of parameters by sharing the $A_i$ across all layers and representing the $B_i$ as independent low-rank factors for each layer.
> > > > > > > > > > >
> > > > > > > > > > > Our approach starts with a three-dimensional representation of the attention weights, sized $d_{\text{model}} \times d \times n$, where $n$ is the number of heads, and $d=d_{model} / n$. This approach exploits the relationships between attention heads to improve parameter efficiency. We have explored different types of updates within the self-attention mechanism and proposed two variants of our FTN: Q&V and O (explained in the Technical Details section of the main paper). The FTN output projection (O) requires ~2x fewer parameters than KAdaptation while achieving comparable performance (average accuracy: 85.4 for KAdaptation and 85.0 for the FTN output projection). Additionally, we demonstrated the flexibility of our method by easily extending it to convolution weights, which are expressed as four-dimensional tensors.

---

### Comment · Reviewer_92f4 · 2024-04-23
**No bandwidth to review at the moment**

Hi, I unfortunately do not have bandwidth to review at the moment since I am preparing for my thesis defense.
Best,
Lucio

---

### Decision · Action_Editor_sxPY · 2024-07-30

**Recommendation:** Reject

**Comment:**

This paper introduces a method called FTN for multi-task and/or multi-domain learning, aiming to maintain accuracy while introducing little additional parameters; the paper uses architectures of convolution and transformer networks to showcase its efficacy.

Three reviewers give thorough and constructive comments. The authors responded and revised the paper accordingly. However, two of the reviewers expect that improved performance would be achieved given the use of additional parameters in FTN; however, this is not well supported by the reported experiments. They also think that the core idea of the paper is not fundamentally different from the LoRA and KAdaptation methods. While the third reviewer suggests an acceptance of the paper, he/she also agrees that new technical contributions of the paper are limited when compared with LoRA. Note that novelty is not a factor for TMLR, but the core idea will gain a limited audience due to insufficient new insights compared to what is known about LoRA and KAdaptation.

**Audience:**

Concerns of limited audience may exist, due to the less insights from this work.

**Claims And Evidence:**

Not well supported. Most of the reviewers expect that by introducing additional parameters into FTN for multi-domain and/or multi-task learning, the results should be better.  The reported results are not-intuitive and not well-justified.